# Cerebral blood volume sensitive layer-fMRI in the human auditory cortex at 7T: Challenges and capabilities

**Lonike K. Faes**[1]*, **Federico De Martino**[1,2‡], **Laurentius (Renzo) Huber**[1‡]

**1** Faculty of Psychology and Neuroscience, Department of Cognitive Neuroscience, Maastricht University, Maastricht, The Netherlands, **2** Department of Radiology, Center for Magnetic Resonance Research, University of Minnesota, Minneapolis, Minnesota, United States of America

‡ FDM and LH are Joint Senior Authors
* l.faes@maastrichtuniversity.nl

**Data Availability Statement:** The anonymized raw data of this study is available and can be downloaded from https://doi.org/10.34894/WCUTUL. The scripts that were used in the

## Abstract

The development of ultra high field fMRI signal readout strategies and contrasts has led to the possibility of imaging the human brain in vivo and non-invasively at increasingly higher spatial resolutions of cortical layers and columns. One emergent layer-fMRI acquisition method with increasing popularity is the cerebral blood volume sensitive sequence named vascular space occupancy (VASO). This approach has been shown to be mostly sensitive to locally-specific changes of laminar microvasculature, without unwanted biases of trans-laminar draining veins. Until now, however, VASO has not been applied in the technically challenging cortical area of the auditory cortex. Here, we describe the main challenges we encountered when developing a VASO protocol for auditory neuroscientific applications and the solutions we have adopted. With the resulting protocol, we present preliminary results of laminar responses to sounds and as a proof of concept for future investigations, we map the topographic representation of frequency preference (tonotopy) in the auditory cortex.

## Introduction

Ultra high field (UHF) magnetic resonance imaging allows the acquisition of functional data with increased sensitivity [1]. This increased sensitivity can be used to breach into the meso-scopic scale in humans [2–9], and the layered functional responses can be leveraged as a proxy for cortical architecture [10].

Gradient-echo blood oxygenation level dependent (GE-BOLD) functional magnetic reso-nance imaging (fMRI) is the conventional approach to collect submillimeter data, due to its relatively high signal-to-noise ratio (SNR) [11]. However, T2*-weighted images collected at 7 Tesla (and higher fields) still contain contributions of both macro- and micro-extravasculature compartments [11,12]. The macrovascular contribution to GE-BOLD originates from both pial vessels and draining vessels that penetrate the cortex orthogonally [13]. This results in two effects: the signal in superficial cortical depths is larger and the layer dependent spatial specific-ity is reduced as activation is drained away from the original locus of neural activity [14–17].

analysis can be found on https://github.com/layerfMRI/repository.

**Funding:** This study was supported by the European Research Council (ERC) under the European Union's Horizon 2020 research and innovation programme (grant agreement No. 101001270) to FDM, authors LKF and FDM were supported by the National Institute for Health grant (RF1MH116978-01), and the Netherlands Organisation for Scientific Research (NWO), NWO VENI grant (016.Veni.198.032) to LH. The funders had no role in study design, data collection and analysis, decision to publish, or preparation of the manuscript.

**Competing interests:** The authors have declared that no competing interests exist.

Regardless, the increased sensitivity, coverage and temporal efficiency of GE-BOLD makes it the most common approach for laminar fMRI (for a recent review see e.g. [18]), also when considering auditory studies [19–23].

While draining effects in GE-BOLD can be reduced with modeling and analyses approaches (see e.g. [24,25]), alternative acquisitions have been proposed to minimize the contribution of macrovasculature. For example, spin-echo (SE) echo planar imaging (EPI) has been used to collect T2-weighted functional data [7,8]. To retain T2-weighted specificity, these applications used segmented EPI acquisitions, while non segmented acquisitions introduce unwanted $T2^*$ contributions [26]. 3D gradient-echo and spin-echo (3D-GRASE) [27,28], has also been used to investigate human laminar and columnar function in both visual and auditory cortices [2,9,29–31]. However, the limited field of view (FOV) of early 3D-GRASE approaches has allowed only the investigation of small portions of cortex and, in auditory studies in particular, often in a single hemisphere ([2,30]; for a review see [32]). More recent 3D-GRASE advancements can mitigate FOV constraints [33]. Furthermore, a large spectrum of alternative approaches is currently under development to optimize the sensitivity and specificity of layer-fMRI experiments [34–44].

Cerebral blood volume (CBV) based imaging is one of the approaches to collect functional data with high spatial specificity. The most commonly used approach to measure functional CBV changes is vascular space occupancy (VASO) [39,45,46]. Previous studies have acquired CBV functional responses alongside conventional BOLD [47–50]. A concomitant acquisition approach of BOLD and VASO has the potential to combine their complementary aspects and facilitate a more comprehensive understanding of the physiology underlying laminar activation changes. Furthermore, a combined acquisition of BOLD and VASO allows researchers to benefit from cumulative quality metrics of both methods, e.g. a high detection sensitivity (in BOLD compared to VASO) and a high localization specificity (in VASO compared to BOLD). VASO has been used to investigate laminar functional responses in visual [51], motor [3,52], somatosensory [53] and prefrontal [54] cortices.

To date, VASO has not been successfully applied to investigate layer dependent functional responses in the human auditory cortex. Despite its lower power compared to BOLD [55], the use of VASO has proven useful outside of auditory cortical areas [3,51–54] and this warrants the need for developing an effective VASO protocol for auditory neuroimaging. Here, we present the results of the exploration of a wide parameter space aimed at mitigating methodological and physiological challenges encountered when using VASO to image the auditory cortex at submillimeter resolution. We evaluated functional images collected at 7T using concurrent measurements of GE-BOLD and VASO. Specifically, we investigated the difference between a 2D- and a 3D-EPI readout and their stability across several participants. The resulting 3D-EPI protocol was then also tested for stability of responses within an extensive session with one volunteer. Lastly, we present preliminary results for laminar profiles of VASO data, and the use of VASO for auditory neuroscience applications by characterizing VASO acquisitions of cortical sound frequency preference (i.e. tonotopic maps).

## Materials and methods

### Ethics

The scanning procedures were approved by the Ethics Review Committee for Psychology and Neuroscience (ERCPN) at Maastricht University, following the principles expressed in the Declaration of Helsinki. Informed consent was obtained from all participants.

## Participants

Participants were healthy volunteers with normal hearing and no history of hearing or neurological disorders. Participants were excluded if they had any standard MRI contraindications (e.g. any metal implants etc.).

Eleven healthy volunteers participated in three separate studies. In study 1 (N = 4), we addressed challenges we encountered in the development of our VASO protocol. In study 2 (N = 5), we evaluated the stability of the protocol with 2D and 3D readouts in four volunteers. Additionally, in one volunteer we investigated the stability of responses with the 3D-EPI. In study 3 (N = 2), we applied the resulting 3D protocol for tonotopic mapping as a proof of principle.

## Scanner

Scanning was performed on a MAGNETOM "classic" 7T scanner (Siemens Healthineers) hosted by Scannexus (Maastricht) equipped with a 32-channel Nova Head Coil (Nova Medical, Wilmington, MA, USA). Sequences were implemented using the vendor provided IDEA environment (VB17A-UHF). We used an in-house developed $3^{rd}$ order $B_0$-shim system (Scannexus) that depends on the vendor provided "3rdOrder ShimSet" feature.

## Auditory stimulation

Sounds were presented to participants in the MRI scanner using MRI compatible ear buds of Sensimetrics Corporation (www.sens.com).

## Slice-saturation slab-inversion VASO

We used a slice-saturation slab-inversion VASO (SS-SI-VASO—[46]) acquisition with either a 3D-EPI [56] or 2D-EPI readout [57]. VASO uses an inversion recovery pulse to effectively null the contribution from the blood magnetization [39,45]. For all of the tested protocols, the inversion delay (i.e. the dead time between the inversion pulse and the VASO signal readout module) was chosen to have the readout block roughly centered around the expected blood nulling time. In SS-SI-VASO, VASO and BOLD images are acquired in an interleaved fashion, which allows for a straightforward combination of the two datasets.

## Reconstruction

The reconstruction of the data was conducted as described in previous studies for SMS-VASO [57] and 3D-EPI VASO [58], respectively. In short, the vendor's in-plane GRAPPA [59] reconstruction algorithms were applied using a 3 × 2 (read direction x phase direction) kernel. Partial Fourier reconstruction [60] was done with the projection onto convex sets (POCS) algorithm [61] with 8 iterations. Finally, the complex coil images were combined using the vendor's implementation of sum-of-squares.

SMS unaliasing was performed on-line on the scanner using a combination of the vendor software and the SMS reconstruction as distributed with the MGH blipped-CAIPI C2P (http://www.nmr.mgh.harvard.edu/software/c2p/sms). SMS signals were first un-aliased with an implementation of SplitSlice-GRAPPA with LeakBlock [62] and a 3 × 3 SliceGRAPPA kernel before entering in-plane reconstruction as described above.

The 3D-EPI reconstruction was based on a previous 3D-EPI implementation [56] using a combination of standard scanner software and a vendor-provided work-in-progress implementation of GRAPPA CAIPIRINHA (Siemens software identifier: IcePAT WIP 571).

## Study 1: Protocol development in pilot experiments

We aimed at implementing and testing a VASO protocol for the auditory cortex that can mitigate a series of methodological challenges. The purpose of study 1 was to explore the protocol parameter space of previously described 2D and 3D VASO sequences with respect to temporal signal-to-noise ratio (tSNR) and minimal artifact level in auditory cortical regions. The protocol resulting from this study will then be subject to quantitative investigations and validations in a subsequent study (study 2). Often, results of these pilot experiments are not reported in manuscripts. However, after encountering several artifacts, we decided to describe the rationale behind the steps we have taken in our mitigation strategies. This might benefit other researchers that encounter similar artifacts and can potentially use the same (or similar strategies) to mitigate their own artifacts.

While developing a protocol we encountered several artifacts of a physiological nature and had to consider different readout strategies. To mitigate physiological noise artifacts, across several sessions, we explored the effect of using a phase-skipped adiabatic inversion pulse with B1-independent partial inversion (based on shapes of a TR-FOCI pulse—[64]) and looked at several readout times (700 ms and 1235 ms) and strategies (2D using multiband and different GRAPPA reference acquisition schemes, and 3D acquisitions).

During the development stages, we tested the protocols for activation elicited by sound presentation. We presented pure tones (800 ms) within the inherent 900 ms dead time of the SS-SI-VASO sequence. The choice of this approach stems from the fact that in auditory fMRI studies, sounds are generally presented inside the silent gap between volume acquisitions (sparse design) [63]. However, this approach resulted in weak auditory evoked fMRI responses in the VASO (and simultaneously acquired BOLD) data. A possible reason for this reduced effectiveness of the sparse design is the relatively short duration of the gap and sound (900 ms and 800 ms respectively) compared to the noise of the BOLD/VASO acquisition time (~2.5 seconds depending on the protocol) [64]. Following this rationale, in studies 2 and 3 we continuously presented auditory stimuli (e.g. the auditory stimulation overlapped with the scanner noise) and played them loud enough to be audible compared to the scanner noise. This approach resulted in larger evoked responses (see results study 2 and 3).

## Study 2: 2D versus 3D comparison

With the resulting protocol of study 1 (see results for an explanation of why specific parameters were selected), we collected two datasets of both BOLD and VASO (0.9 mm isotropic and 12 slices), one with a 2D readout (TR = 1833.5 ms; TE = 21 ms; flip angle = 70˚; GRAPPA = 3; reference scan = segmented) and one with a 3D readout (TR = 1609 ms; TE = 22 ms; variable flip angles between 16˚(first segment of readout block) and 30˚ (last segment of readout block); GRAPPA = 3; reference scan = FLASH [65]) in four volunteers.

Participants were asked to passively listen to a series of sounds consisting of multi-frequency sweeps. Stimuli were presented following a blocked design with 20 volumes of sound stimulation followed by 20 volumes of rest. Each run consisted of thirteen stimulation blocks lasting about 11 minutes. A recording of the stimuli is available here: https://layerfmri.page.link/aud_stim. In each participant we collected two runs (S1 and S4 Figs) or 3 runs (S2 and S3 Figs) with a 2D readout and a 3D readout.

To further test the reliability of the 3D acquisition protocol (see results for why we opted for a 3D readout), we scanned one additional volunteer in an extensive session in which we collected 12 runs with the same block design explained above to measure stability of responses across independent splits of the data. We opted for scanning this volunteer in one long session, instead of scanning the same participant twice in two shorter sessions, because alignment across sessions with our current experimental setup is challenging due to the small coverage.

### Study 3: Tonotopy

Simultaneous BOLD and VASO data were collected using the 3D sequence (after finalizing study 2) described above (0.9 mm isotropic; 12 slices; TR = 1609 ms; TE = 22 ms; GRAPPA = 3; reference scan = FLASH), variable flip angles between 16˚ (first segment of readout block) and 30˚ (last segment of readout block). In addition, we collected anatomical data (with optimized gray/white matter contrast) using MP2RAGE (TR = 6000 ms, TE = 2.39 ms, TI1/TI2 = 800/2750 ms, FA1/FA2 = 4˚/5˚, GRAPPA = 3 and 256 slices) [66] at a resolution of 0.7 mm isotropic.

Participants passively listened to tones varying slightly around 7 different center frequencies (130, 246.2, 466.3, 883.2, 1673, 3168 and 6000 Hz). Center frequencies were presented following a blocked design. Stimulation blocks (23 seconds) contained forty-six tones (500 ms each) varying 0.2 octaves around the center frequency. Each stimulation block was followed by a rest period (23 seconds). Functional runs consisted of fourteen stimulation blocks with a total duration of approximately 11 minutes per run. In one participant we collected four runs and in a second participant five runs. Before each tonotopic experiment tones were equalized for perceived loudness.

### Functional data analysis

Preprocessing in all studies was done in the same way. All functional images were sorted by contrast, resulting in a (BOLD-contaminated) VASO and a BOLD time series. The first three volumes of each time series were removed to account for the steady state. Each time series was motion corrected using SPM12 (Functional Imaging Laboratory, University College London, UK). The estimation of the motion parameters was restricted to a mask of the temporal lobe. Next, the time series were temporally upsampled by a factor of 2. This resulted in an interpolated TR of 1.15 seconds in the 3D readout and about 1.3 seconds in the 2D readout. As in previous studies, we corrected for the BOLD contamination in the VASO data using the open software suite LayNii (version 2.2.0) [67].

In study 2, activation maps were created using AFNI (3dDeconvolve—version 21.2.04 and Matlab). We used a General Linear Model and normalized the time course to z-scores (when comparing 2D versus 3D) and to percent signal change (to test the reliability of the 3D protocol). The resulting F-maps portray normalized differences between periods of auditory stimulation and rest. Two-dimensional ROI's were drawn manually in spatially upsampled EPI space and were divided in 7 equivolume layers [68] with which layer plots were created using LayNii.

In study 3, after preprocessing, functional data were first aligned to the anatomical data using Brainvoyager (version 22.2—Brain Innovation, Maastricht, The Netherlands). Anatomical images were processed in BrainVoyager (BV). We used an automatic segmentation pipeline of BV with which we created a mid gray matter surface. For statistical analysis we used a General Linear Model with one predictor for each center frequency. Time series were normalized to percent signal change prior to statistical analysis. Tonotopic maps were created using best frequency mapping [69] and were interpolated across depths and projected on the mid gray matter surface.

## Results

### Study 1: Protocol development

In study 1, we aimed to mitigate several methodological and physiological challenges that we encountered while exploring the use of different parameters. In the following section, we will

discuss the rationale behind the use of specific parameters and how they have helped reduce the artifacts in our data.

First, compared to other cortical areas, the auditory cortex has an exceptionally short arterial arrival time of approximately 0.5–0.8s (see reference [70], in particular the results reported in its Fig 7B). This is approximately 1-2s earlier than the primary visual cortex. Such short arterial arrival times can result in the unwanted inflow of fresh (uninverted) blood during the VASO readout. The inflow effects result in very bright vessels in both the BOLD and the VASO data. However, the ratio between the background signal and the signal from the vessels is higher in the BOLD data compared to the VASO data. This results in lower relative contrast between tissue types in the VASO data compared to the BOLD data (Fig 1A). To mitigate this challenge, we explored the usage of a phase-skipped adiabatic inversion pulse with B1-independent partial inversion (based on shapes of a TR-FOCI pulse—[71]) that minimized these contaminants at the cost of SNR. Reducing the inversion efficiency by means of the phase skipped adiabatic inversion pulse can reduce the blood nulling time so that it is shorter than the arterial arrival time, mitigating inflow artifacts. Depending on the TR, the inversion efficiency and excitation flip angles that are used, the tissue signal can be reduced by about 30%.

Second, we explored the effect of readout time (and its relationship to the cardiac cycle) on VASO data in the temporal cortex. Initially, we used a readout time of about 1235 ms (in one participant), which is longer than the cardiac cycle. Such a long readout time resulted in loss of contrast around Heschl's gyrus (HG). Therefore, we opted for using a readout time shorter than the cardiac cycle (700 ms) in subsequent volunteers in all three studies, to mitigate this artifact. An additional independent component analysis (FSL MELODIC, 30 components) on the VASO time series (Fig 1B) collected with a readout longer than the cardiac cycle showed that the component with the largest variance was a typical vascular artifact centered on the large vessels in the auditory cortex. These two results exemplify the effect of physiological noise originating from the cardiac cycle after which we decided to shorten the readout time.

These physiological noise artifacts additionally made us consider different readout strategies. High-resolution VASO is commonly used in combination with a 3D signal readout (e.g. 3D-EPI). However, since the auditory cortex, especially the medial portion of HG, is located right next to large feeding arteries, the partitioned 3D-EPI approach can result in higher susceptibility to physiological noise. To compare it to a 2D-EPI readout (study 2), the optimization of parameters specific to the 2D readout was required. In particular, the location of the auditory cortex requires large in-plane imaging FOVs, resulting in a large matrix size, and low bandwidth in the phase encoding direction for submillimeter acquisition protocols. The correspondingly long readout duration makes the acquisition protocol more susceptible to Generalized Autocalibrating Partially Parallel Acquisitions (GRAPPA) [59] artifacts. To find an effective protocol we compared the tSNR over 40 volumes resulting from an SS-SI-VASO acquisition with 2D readout at 0.9 mm isotropic employing different GRAPPA references: single-shot, segmented and FLEET [16] with three different flip angles (2, 30 and 90 degrees) (Fig 1C). We calculated the ghost level by taking the ratio of intensity values of a region outside of the brain and one region centered on the auditory cortex. We expressed this as a percentage value that can be found in the first row of Fig 1C. The FLEET ACS approach exhibited worse ghosting compared to single-shot and segmented in our experimental setup. Therefore we decided to refrain from using FLEET in the following experiments. Since the echo time and the phase evolution of single-shot GRAPPA reference lines are approximately three times longer/stronger for single-shot ACS compared to the segmented approach, we expected that using a segmented approach would mitigate intermittent ghosting across the time series. This is expected to result in stable tSNR values across protocols and participants. Thus, we decided to

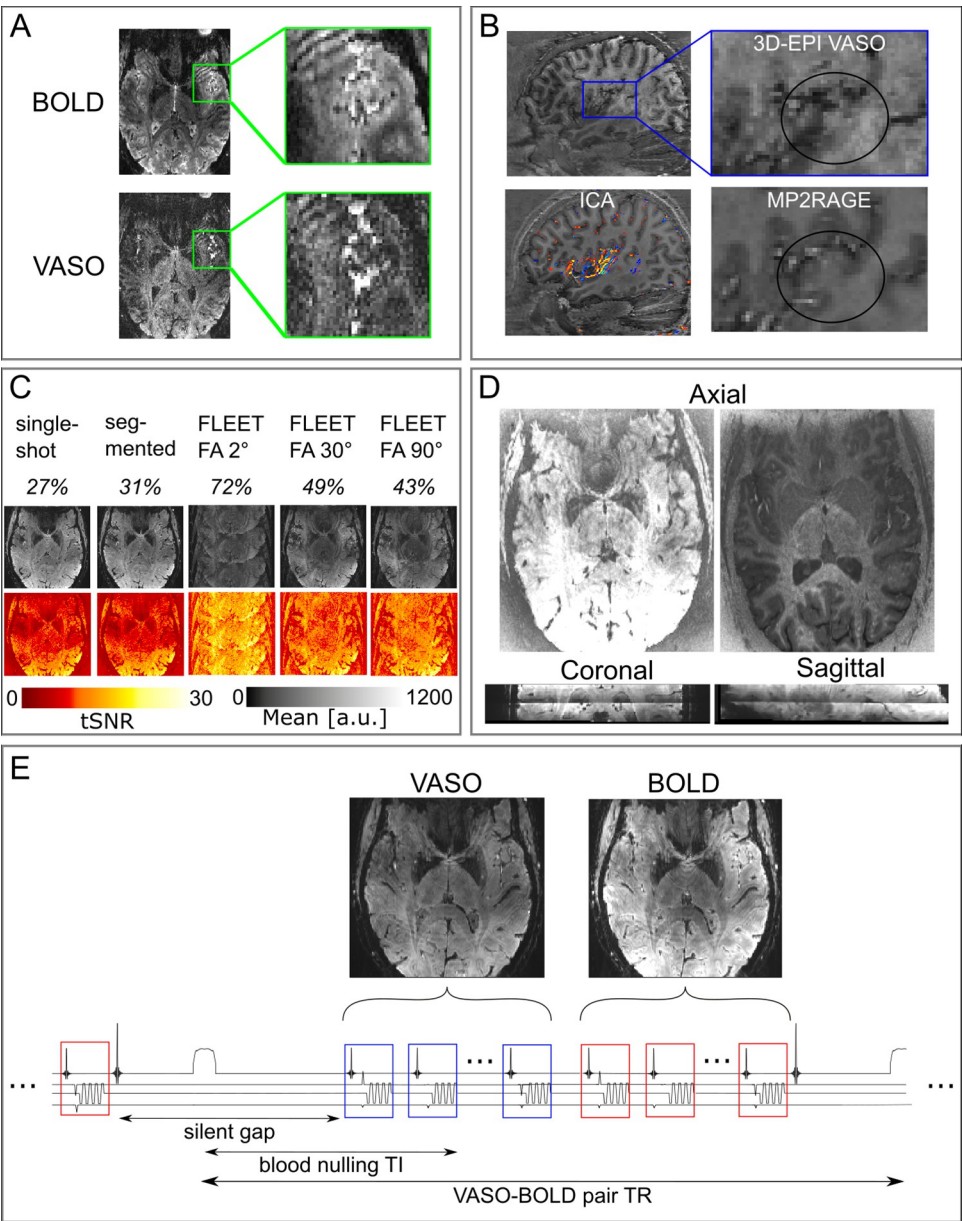

**Fig 1. Overview of the challenges encountered when acquiring VASO data in the auditory cortex.** (A) Inflow effects were found in both GE-BOLD and VASO in temporal regions. However, the VASO signal seemed to be more affected by the inflow of not-nulled blood. (B) Cardiac pulsation effects reduced image contrast due to long 3D-EPI readouts. In the functional images, the contrast in our region of interest seemed to be particularly affected. Additional ICA analysis (left bottom) showed the main components around Heschl's gyrus. (C) In the presence of physiological noise, there is a tradeoff in the amount of ghosts and the tSNR when evaluating different GRAPPA auto calibration signal (ACS) acquisitions. The first row contains the percentage of background signal compared to signal in the auditory cortex. The second row gives an impression of the ghost level and the third row gives an illustration of the tSNR. These tests were conducted for the protocol with 2D-SMS readouts. (D) 2D-SMS VASO resulted in T1-weighted slice-wise intensity differences that were most visible in the middle of the slab. The two axial slices show the intensity differences between two "consecutive" slices (with the same signal intensity scaling). (E) Schematic depiction of one TR of the final SS-SI-VASO sequence. An inaudible phase-skipped adiabatic pulse is used in the inherent silent gap of this sequence. This is followed by the acquisition of a volume of VASO and a volume of BOLD.

use the segmented approach for the remainder of the study as this is expected to be the best compromise between artifact level and tSNR in temporal areas.

Finally, we considered the use of 2D simultaneous multi slice (SMS—also known as multi-band) [72,73] EPI readouts in VASO in order to 'freeze' cardiac-induced vessel pulsation artifacts. The use of SMS results in different effective inversion times across slices and in our investigations this translated to sudden jumps of signal intensity in the VASO data (Fig 1D). As this complicates the performance of retrospective motion correction and results in spatially heterogeneous tSNR we did not use SMS in the comparison in study 2.

A schematic depiction of the final protocol is illustrated in Fig 1E (and a complete parameter list is made available here: https://github.com/layerfMRI/Sequence_Github/tree/master/Auditory). In particular, we used an (inherently) inaudible adiabatic inversion pulse with a 30 degree phase skip, a readout time of 700 ms (which is shorter than the cardiac cycle) and a 70 degree reset-pulse [74] at the end of each acquisition of a VASO-BOLD pair. The purpose of the reset pulse was to effectively saturate stationary $M_z$-magnetization of cerebrospinal fluid (CSF) and gray matter (GM) before the application of the consecutive inversion pulse. At the blood nulling time in the subsequent TR, this results in a positive $M_z$-magnetization of CSF with a magnitude smaller than GM. Having a positive CSF $M_z$-magnetization in SS-SI-VASO is in contrast to the negative CSF magnetization in the traditional VASO approach. The suppressed CSF signal (see contrast in Fig 1E) mitigates potential biases of dynamic CSF volume changes that have previously been reported to impose a source of bias for VASO applications in the auditory cortex [75]. The effective temporal resolution was 2.3 seconds.

## Study 2: 3D-EPI versus 2D-EPI

The presentation of auditory stimuli resulted in reliable responses in the bilateral auditory cortex for VASO (except for participant 2 in the 2D readout acquisition—see Fig 2A) and for BOLD (S1 Fig). For VASO, the 3D readout resulted in higher z-scores in bilateral auditory cortex, while this benefit was not directly visible in the BOLD data at these resolutions (S1 Fig). Even though the activation scores in VASO are relatively weak, they are within the expected regime of sub-millimeter protocols [58]. These results are somewhat consistent with previous 2D vs. 3D comparisons of VASO in the primary motor cortex [76]. Here we extend these findings for the physiological-noise constrained primary auditory cortex.

Average time courses of active voxels calculated in percent signal change are plotted in Fig 2B against our experimental paradigm (yellow bars). These time courses exemplify the negative percent signal change of VASO following auditory stimulation.

Since the VASO signal is a composite signal from blood-nulled and not-nulled (BOLD) images, its detection sensitivity is indirectly dependent on the noise level of BOLD too. We believe the result that VASO benefits from 3D-EPI more strongly than BOLD, is thus mostly driven by the relatively lower tSNR of blood-nulled images compared to non-nulled BOLD images.

## Cortical depth-dependent responses

Fig 3 shows the layer profiles obtained in 2D regions of interest (ROIs; covering Heschl's Gyrus [HG]). In VASO, the signal had a tendency to increase within gray matter. However, the cortical depth dependent signal also showed a reduction at the pial surface (CSF/GM in Figs 3 and 4), indicating its reduced sensitivity to macrovasculature. Separate analysis on the BOLD data using the same ROI definition, showed a monotonic increase in functional activation towards the cortical surface (S2 Fig) and no decrease on the pial surface. Similar results were obtained when defining ROIs based on functional activation (response to sounds) in the

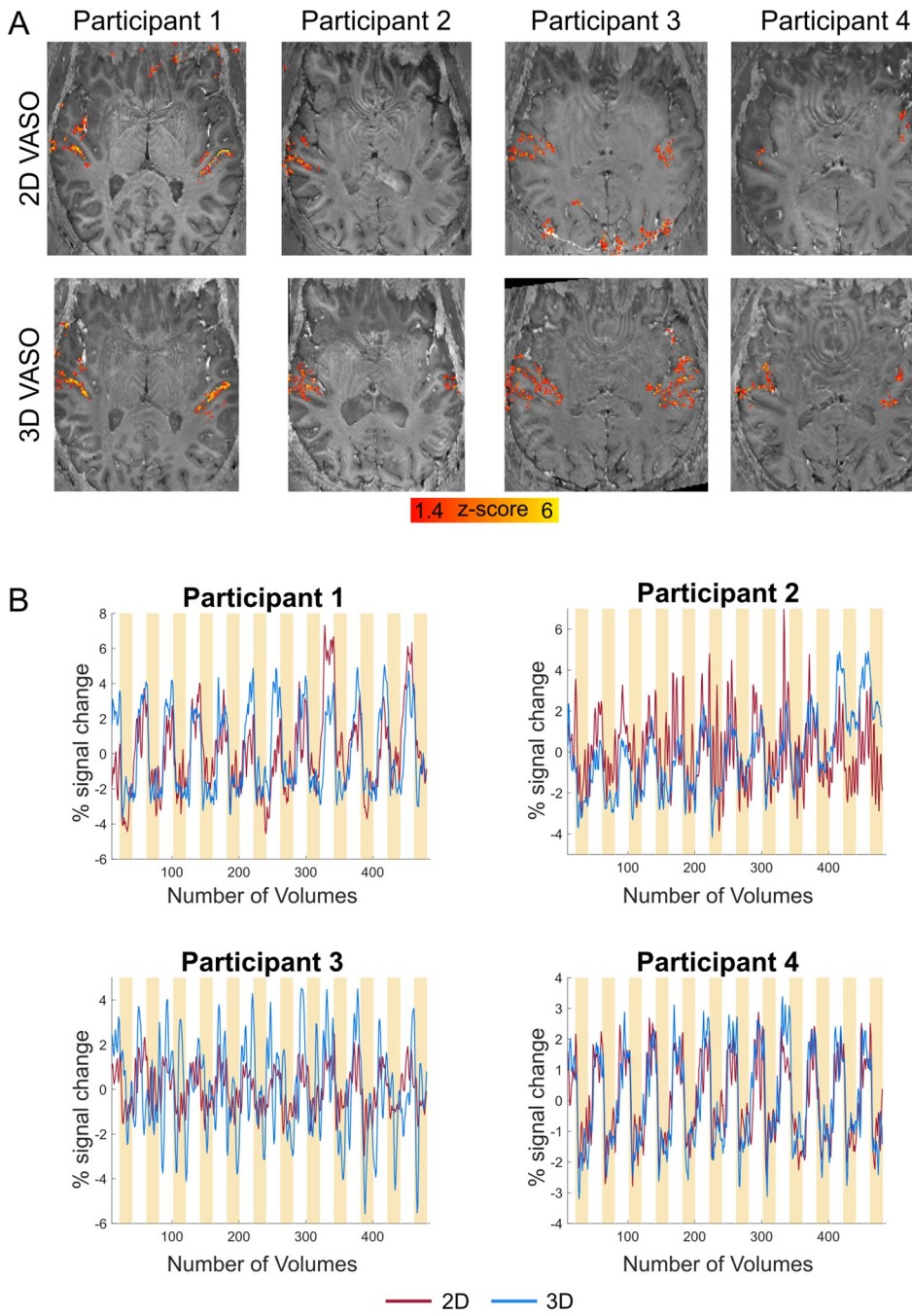

**Fig 2. Activation maps and time courses of VASO.** (A) Z-scored activation maps overlayed on distortion corrected mean EPI images (per participant and readout). For our data, using a 3D-EPI readout seems to be beneficial in VASO. (B) VASO time courses (average coming from active voxels) calculated in percent signal change illustrate the negative percent signal change when auditory stimuli are presented. Yellow bars indicate the presentation of auditory stimuli.

medial anterior part of HG on the left hemisphere (Fig 4 for VASO and S3 Fig for BOLD). Both the activation maps and the laminar analysis indicate that a 3D readout is beneficial for collecting VASO data (higher z-scores and increased reliability).

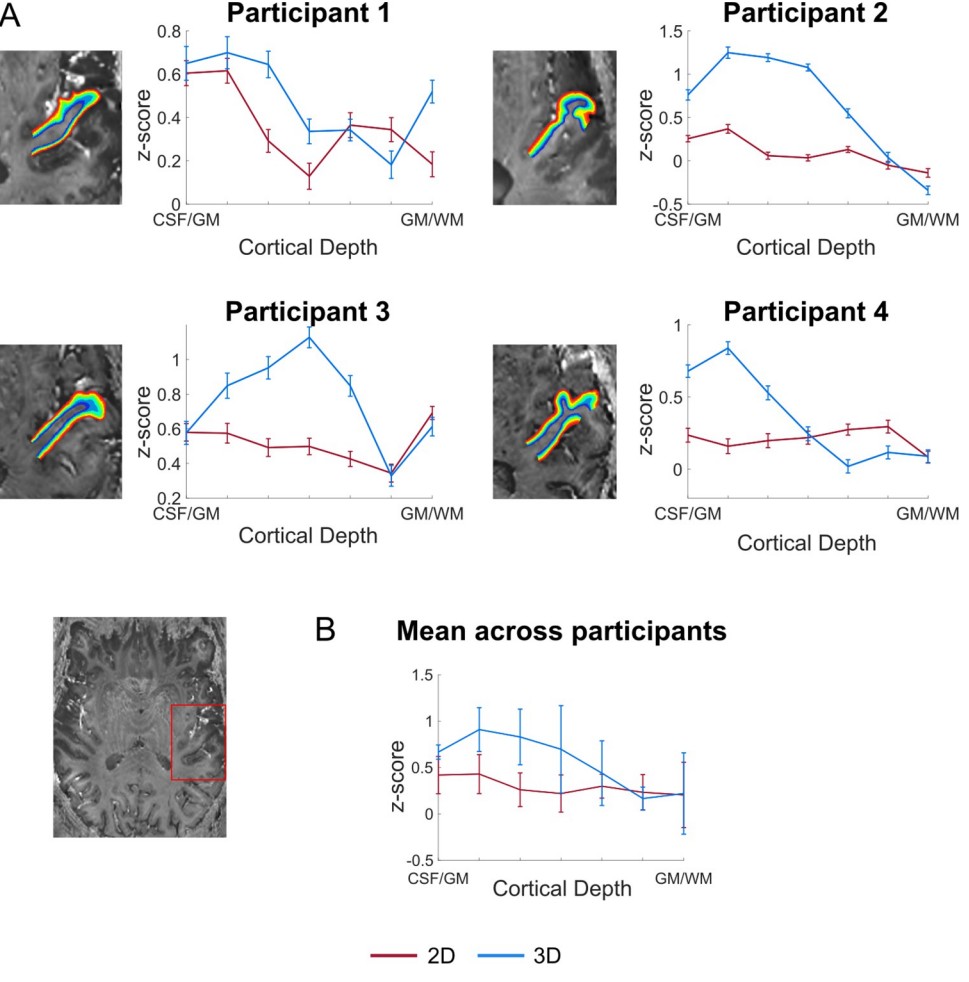

**Fig 3. Z-scored cortical depth-dependent activation changes for the 2D- and 3D-EPI VASO data.** (A) The anatomically-informed ROI was drawn on the bias field corrected mean 3D-EPI VASO (example at the left bottom). The fMRI layer-dependent changes across depths for each participant. (B) Average z-scored layer-dependent activation changes across participants.

## Reliability of responses

To measure the reliability of auditory responses in our 3D acquisition protocol, we collected 12 runs with the 3D acquisition protocol in one volunteer. By analyzing two independent splits of the data (6 runs each) we evaluated the stability of the responses (Fig 5). Whole brain GLM's were computed for each split independently. In Fig 5A, we show the F-maps thresholded at p<0.05 uncorrected (and a minimum cluster size of 10 voxels) for both splits to illustrate the spatial reliability of activation maps obtained across splits. The time course of active voxels (in percent signal change, Fig 5B) demonstrates the expected negative responses upon stimulation. Compared to Fig 2B, time courses in Fig 5B are less noisy, illustrating the benefit of averaging over more runs.

To investigate the reliability of laminar responses, we also present the laminar activation plots for the two independent data splits (Fig 5C) extracted from an ROI (drawn based on functional activation on an axial slice).

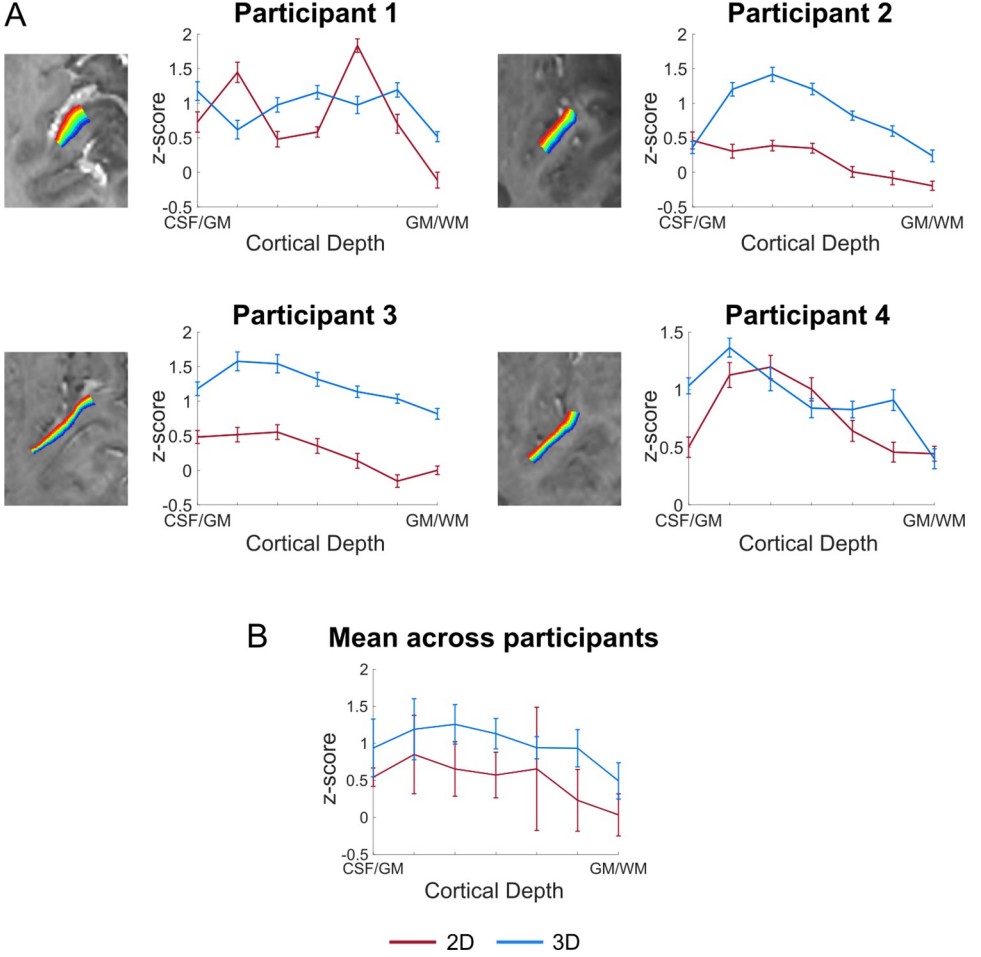

**Fig 4. Z-scored cortical depth-dependent activation changes for the 2D- and 3D-EPI VASO data.** The ROI was drawn on an axial slice as shown in Fig 3A and was based on functional activation. (A) The fMRI layer-dependent changes across depths for each participant. (B) Average z-scored layer-dependent activation changes across participants.

## Study 3: Tonotopic maps

In study 3, the presentation of pure tones resulted in responses in the bilateral auditory cortex for both BOLD and VASO. Mid-gray matter anatomical surfaces were created from a WM/GM segmentation and inflated (Fig 6) to visualize HG (outlined in black) and the planum temporale/polare. The analysis was confined to voxels showing both a positive signal change for BOLD (at a threshold of p<0.05 uncorrected) and a negative VASO signal change. Tonotopic maps (Fig 6) show the expected high-low-high frequency gradient along HG in VASO (see e.g. [77] for a comprehensive discussion on the expected topography of tonotopic maps). The same gradient is present in the BOLD data (S4 Fig) as shown in previous studies using GE-BOLD [77].

## Discussion

Despite the fact that layer-fMRI VASO can provide valuable information in sub-millimeter and layer-fMRI applications [3,51–54], it has not been successfully applied in the human auditory cortex. This is in contrast to GE-BOLD, which has been used to image laminar and

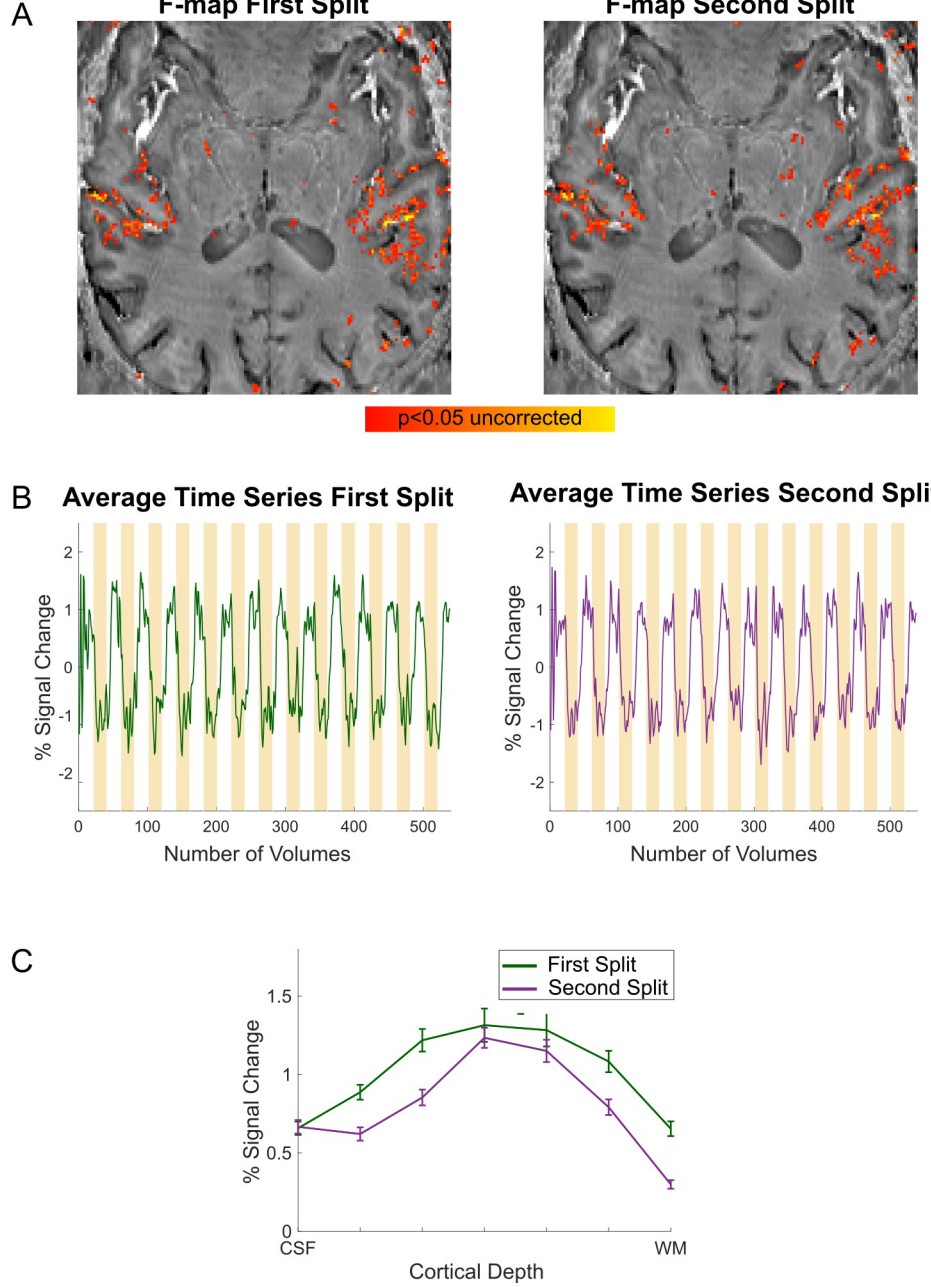

**Fig 5. Stability of responses.** (A) F-maps (p<0.05 uncorrected) are displayed on a bias field corrected mean EPI VASO image. (B) Time courses in percent signal change are displayed against our experimental paradigm, illustrating the negative signal change upon auditory stimulation (yellow bars). (C) Laminar responses in a functional activation based ROI for both splits of the data.

columnar responses in the temporal lobe [2,19–21,23]. In this study, we aimed to develop a VASO protocol for laminar fMRI investigations of the auditory cortices by mitigating methodological and physiological challenges.

Starting from a protocol that was previously successfully used [51], the location and vascular physiology of the auditory cortex resulted in several artifacts. This required us to reconsider acquisition parameters and approaches that have helped to improve layer-fMRI applications

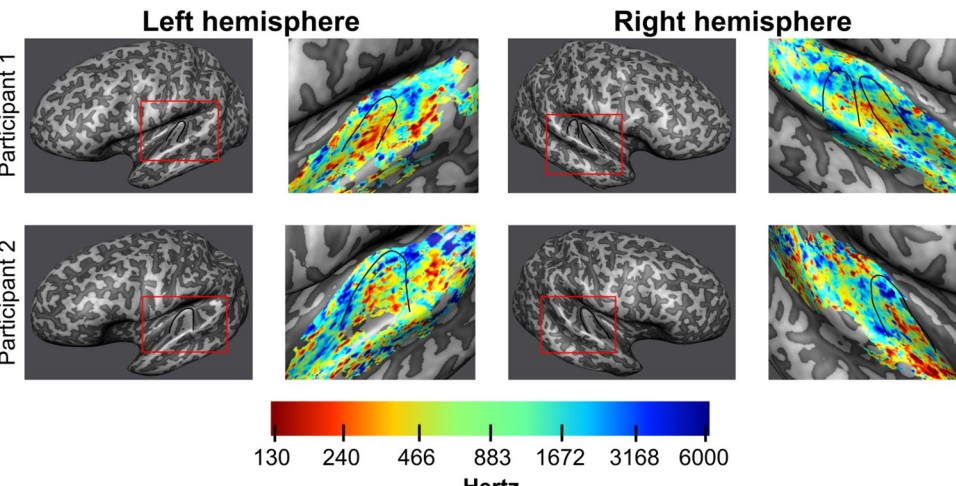

**Fig 6. Tonotopic maps VASO.** Inflated mid-gray matter surface meshes were created to visualize tonotopic maps coming from the VASO data. Red boxes outline the part of the mesh from which the tonotopic data was sampled. Heschl's gyrus is outlined in black. On the right of each inflated surface, tonotopic maps are displayed for both hemispheres of the two participants. The expected tonotopic high-low-high frequency preference gradient is respected.

but whose proof of generalizability across brain areas is still limited. The need to account for the specific vascular physiology of the auditory cortex, resulted in the use of a readout time shorter than the cardiac cycle and optimization of the inversion pulse (Fig 1). To evaluate possible physiological contamination when using the standard 3D readout in VASO, we considered the use of a 2D readout. To develop an efficient 2D protocol for VASO, we combined techniques (such as FLEET for GRAPPA reconstruction and SMS acquisition) which are often used (in auditory neuroscience studies) when collecting submillimeter GE-BOLD data. To our surprise, while these approaches showed the expected utility in the GE-BOLD data, they did not result in the expected increase in sensitivity when considering the VASO data (Fig 1). Study 1 resulted in two protocols (2D and 3D readout) for VASO fMRI in the temporal lobe.

A few words are warranted on the possible influence of CSF volume changes. Assuming the skull as a container of a fixed volume, increase of one compartment, e.g., CBV, must be compensated by volume decrease in another compartment, e.g., GM or CSF. VASO signal change is based on the idea that CBV increase is compensated by GM volume decrease only. However, depending on the brain region stimulated, a small dynamic change in CSF volume in the range of 0.5% (for neurally-induced tasks) to 10% (for systemic gas-breathing induced hypercapnia) has been experimentally observed [75,78]. Such stimulus-dependent variations in CSF volume could cause an incorrect calculation of CBV changes from the VASO signal change [75,78,79]. In contrast with to CSF-nulled ACDC VASO [80] and VASO FLAIR [78] techniques, the SS-SI VASO signal changes in this study (with the employment of a 70 deg spin-reset pulse) reflect a positive CSF z-magnetization. Thus, the CBV change presented here reflects both components of the CBV change—the CBV increase that is compensated by a GM volume decrease as well as the CBV increase that is compensated by CSF volume decrease—with similar weighting.

The comparison of the 2D and 3D protocols resulting from study 1 (study 2) showed an increased stability and SNR when using the 3D-EPI readout, despite its susceptibility to physiological noise. Note that the benefit for the 3D readout was particularly visible in the VASO time series (and not the BOLD data). It has been previously shown that the superiority of 2D-SMS or 3D-EPI readout strategies at 7T in conventional BOLD is highly dependent on and

specific to the acquisition and analysis details including the TR, acceleration factor, resolution, physiological noise correction and number of slices [76,81–84]. The BOLD results presented here are in agreement with this literature (see S1–S3 Figs).

We examined laminar profiles of activation elicited by the sounds presented in study 2. Similarly to previous studies investigating the specificity of laminar functional responses in auditory cortex [30] (using 3D-GRASE), we did not observe a clear peak in functional response in middle cortical depths in the 2D versus 3D comparison. Firstly, this could be due to limited power for the data reported in Figs 3 and 4 (2 or 3 runs depending on the volunteer). A larger data sample (6 runs) resulted in reproducible (across independent splits) laminar profiles with a more pronounced peak in middle gray matter (Fig 5C). Nevertheless the variability in laminar profiles we observed in study 2 could also be caused by the nature of the stimulation and analysis steps. As the auditory stimuli in study 2 were composed of complex dynamic sounds presented for about 20 seconds, it is unclear what the expected neural laminar profile would be in absence of any control for attention or another task. Second, we defined regions of interest for the laminar profiles based on macro-anatomy (anterior HG) or activation. The effect that this has on sampling the laminar activation profiles in auditory regions, whose cytoarchitecture overlaps only partly with macro-anatomical features (see e.g. [85]), is beyond the scope of this paper but could be an interesting venue for future investigations. What we did observe was that while the signal in the upper layers has the tendency to be larger than in middle and deeper layers, the signal decreases again at the pial surfaces. This is expected due to VASO's insensitivity to large pial veins. As expected, GE-BOLD data resulted in an increased response towards superficial layers (S2 and S3 Figs) without a reduction on the pial surface. This profile is characteristic of GE-BOLD submillimeter acquisitions and is resulting from vascular draining and the contribution of large vessels on the cortical surface. If confirmed when analyzing a larger sample, a more controlled stimulus design, and within a more extended portion of temporal areas, the fact that vein-free VASO signal changes [49] within GM decrease as a function of cortical depth, could be interpreted as a validation of previous BOLD results (e.g. indicating that the signal trends visible in the BOLD signal in temporal areas cannot be solely explained by draining vein effects alone). It is important to note that while we here demonstrate that VASO auditory responses are not affected by draining and large vascular contributions on the cortical surface, we do not imply that in presence of careful controls [4,5,30] or with the use of modeling techniques [24,25] GE-BOLD data cannot be used to investigate laminar cortical processing.

To assess the reliability of functional responses collected with our 3D VASO protocol, we measured functional responses in an extended session in one additional volunteer. The resulting data were split into two independent sessions (6 runs each). F-maps, time courses and laminar profiles obtained from the two independent splits indicate that the acquisition (with our 3D VASO protocol) of functional responses in auditory cortical regions is spatially reliable and results in stable temporal responses (with an expected negative response upon stimulation) and reproducible laminar profiles.

With the resulting 3D protocol, as a first proof of concept of the usability of VASO fMRI for the investigation of cortical processing in the temporal lobe, we presented results from a tonotopic experiment. Neurons throughout the auditory pathway display preferential tuning to the sound frequency [86] and using fMRI the topographic arrangement of frequency preference (tonotopy) can be mapped in single individuals [19,69,77,87]. Tonotopy shows a typical topography with a low frequency region residing primarily on the HG and regions preferring high frequency bordering it both posterior medially and anterior laterally (for a description see [77]). This characteristic topography makes tonotopy a possible benchmark for auditory functional acquisitions. The large scale tonotopic gradient covering the superior temporal

plane was visible in the VASO data. This initial promising result opens the venue to further investigations on the specificity of the VASO signal across cortical depths [30].

Despite the shown applicability of VASO for auditory fMRI, we deem it necessary to outline some limitations (many not specific to auditory studies) that require consideration when setting up a neuroscientific (laminar) fMRI study. While VASO is more sensitive to microvascular CBV increases, it is also characterized by a reduced detection sensitivity (as indicated by generally lower z-scores in Figs 2–4 than in S1–S3 Figs). To compensate for this effect a typical approach is to average across runs. As a result, extending averaging across sessions would require careful consideration of approaches for inter session alignment (and placement) of the relatively small slab (12 slices in our case). Future investigations may have to address issues related to detection sensitivity and its dependence on experimental design and sound presentation schemes. In addition, when using VASO, functional runs are typically acquired with an identical design as averaging is performed on the raw time series before BOLD correction to limit noise amplification. This calls for careful balancing of conditions within functional runs. To increase sensitivity we also employed long stimulation periods (block design). Evaluating the sensitivity of event-related functional responses with VASO [88] would increase its usability (e.g. to prediction-error related responses in typical oddball designs). Moreover, alternative approaches for increasing sensitivity such as denoising (e.g. NORDIC—[89]), should be considered in future investigations. Finally, while to compensate for physiological noise effects we decided to use a readout train shorter than the cardiac cycle, in the future it may be interesting to consider higher order physiological noise correction methods in k-space.

We believe that the significance of this work is multi-fold. In study 1, we describe the approach we followed to tackle the main challenges encountered when using VASO for sub-millimeter auditory investigations. Following these steps may prove useful in case the resulting protocol we describe here would not generalize outside of the specific applications (as well as coils and imaging resolutions) we present. Nevertheless, we believe our results represent a first necessary step towards generalization as the protocol resulting from study 2 is made available for the user base of application-focused neuroscientists for testing in a wider range of application settings. The sequence binaries and the importable protocols are publicly available via 'SIEMENS' sequence 'app-store' on TEAMPLAY for any users of a 'classical' MAGNETOM 7T, which is the most widely used 7T scanner version around the world today. Users of other scanner versions and vendors can benefit from this protocol-development study as they can re-implement the acquisition approaches as described in study 1.

A first step towards a generally applicable VASO protocol for auditory neuroscientific studies is furthermore relevant as VASO may allow application studies that are not straightforwardly addressable with the vein-bias of conventional GE-BOLD, such as single-task condition experiments. In such experiments, utilizing VASO protocols (such as the one we developed) alongside with BOLD can be useful to augment the understanding of the neurovascular origin of the fMRI signals. Other example studies, where acquiring VASO and GE-BOLD simultaneously may be beneficial, might be related to research questions of altered vascular baseline physiology (e.g. in studies about pharmacological interventions, aging and surgical interventions). Furthermore, we think that the concomitantly acquired VASO and BOLD data can be useful to calibrate existing layer-fMRI BOLD models [14,24,25,90–93] and extend their applicability across brain areas. For example, future GE-BOLD studies that want to apply venous-deconvolution model-inversion and may not find an increased response in the middle layers, can use the data we present here to increase the confidence in their results. The imaging protocol developed here may have implications beyond the auditory cortex. The auditory cortex is not the only brain area challenged by proximal macro-vessels with substantial physiological noise. There are many other brain areas in which sub-millimeter VASO was not

successfully applied until now, for example, hippocampus, insular cortex, claustrum, entorhinal cortex, and thalamic nuclei. Researchers investigating areas with similar artifacts could test whether following similar strategies might benefit them in these challenging areas. Finally, the main aim of this work was to provide the auditory research community with a viable VASO protocol for laminar fMRI studies, which is now available for testing by the community.

To conclude, our results demonstrate that, when using carefully chosen parameters, VASO can be used to investigate cortical responses in the bilateral temporal cortex. While VASO has a lower detection threshold compared to GE-BOLD, it is believed to be dominated by microvascular CBV increase close to the site of neural activity changes. A combined acquisition approach of BOLD and VASO, as described here, may allow benefitting from the quality features of each method.

## Supporting information

**S1 Fig. Activation maps of BOLD.** Z-scored activation maps overlayed on distortion corrected mean GE-BOLD EPI images (per participant and readout). The color map was chosen to match the VASO data displayed in Fig 2A.
(TIF)

**S2 Fig. Z-scored cortical depth-dependent activation changes for the 2D- and 3D-EPI BOLD data.** (A) Functional layer-dependent changes across depths for each participant. The BOLD data is coming from the same anatomically-based ROI that was used to calculate the layer-dependent VASO changes in Fig 3. (B) Average z-scored layer-dependent activation changes across participants.
(TIF)

**S3 Fig. Z-scored cortical depth-dependent activation changes for the 2D- and 3D-EPI BOLD data.** (A) Functional layer-dependent changes across depths for each participant. The BOLD data is coming from the same functional activation-based ROI that was used to calculate the layer-dependent VASO changes in Fig 4, drawn on an axial slice as shown in Fig 3. (B) Average z-scored layer-dependent activation changes across participants.
(TIF)

**S4 Fig. Tonotopic maps BOLD.** Inflated mid-gray matter surface meshes were created to visualize tonotopic maps created with the BOLD data. On the right of each inflated surface, tonotopic maps are displayed for both hemispheres of the two participants. Heschl's Gyrus is outlined in black. A tonotopic high-low-high frequency preference gradient is visible in the data.
(TIF)

## Acknowledgments

The sequence used here is based on sequence code kindly written and provided by Benedikt Poser. We thank Steve Cauley at MGH for sharing the interface of their image reconstruction for use with the SMS acquisition. We thank Miriam Heynckes for advice on the use of auditory stimulation setups. We thank Omer Faruk Gulban for early contributions to this work. We thank Chris Wiggins for providing the 3rd order shimming tools used here. Scanning was supported by FPN (Faculty of Psychology and Neuroscience) via the MBIC grant scheme.

### Diversity statement

Recent work in several fields of science has identified a bias in citation practices such that papers from women and other minorities are under-cited relative to the number of such

papers in the field [94]. In the human layer-fMRI community the average of the gender citation bias is 84% male, 15% female (https://layerfmri.com/papers/). We obtained the gender of the first author of each reference. By this measure (and excluding self-citations to all authors of our current paper), our references contain 78% male first and 22% female first. This method is limited in that: (i) names, pronouns, and social media profiles used to construct the databases may not, in every case, be indicative of gender identity, and (ii) it cannot account for intersex, non-binary, or transgender people. We look forward to future work that could help us to better understand how to support equitable practices in science.

## Author Contributions

**Conceptualization:** Lonike K. Faes, Federico De Martino, Laurentius (Renzo) Huber.

**Data curation:** Lonike K. Faes.

**Formal analysis:** Lonike K. Faes, Laurentius (Renzo) Huber.

**Funding acquisition:** Federico De Martino.

**Investigation:** Lonike K. Faes, Federico De Martino, Laurentius (Renzo) Huber.

**Methodology:** Lonike K. Faes, Laurentius (Renzo) Huber.

**Project administration:** Lonike K. Faes.

**Supervision:** Federico De Martino.

**Visualization:** Lonike K. Faes, Laurentius (Renzo) Huber.

**Writing – original draft:** Lonike K. Faes, Federico De Martino.

**Writing – review & editing:** Lonike K. Faes, Federico De Martino, Laurentius (Renzo) Huber.

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
