## [Decision Letter · Decision Letter 0]

29 Sep 2022

PONE-D-22-23894Cerebral blood volume sensitive layer-fMRI in the human auditory cortex at 7T: Challenges and capabilitiesPLOS ONE

Dear Dr. Faes,

Thank you for submitting your manuscript to PLOS ONE. After careful consideration, we feel that it has merit but does not fully meet PLOS ONE’s publication criteria as it currently stands. Therefore, we invite you to submit a revised version of the manuscript that addresses the points raised during the review process. Both reviewers felt that the study is very interesting and that it makes a potentially very important contribution. There were, however, also certain concerns, particularly in the comments of Reviewer 1, which would need to be addressed. Some of these concerns might be addressable by textual modifications, by toning down the assertions, more frank discussion of the relative weaknessess, such at the weakness of effect sizes that Reviewer 1 mentions. However, please also consider the suggestions for providing additional data/analysis results to address these concerns.

We look forward to receiving your revised manuscript.

Kind regards,

Jyrki Ahveninen

Academic Editor

PLOS ONE

Journal Requirements:

2. Please upload a copy of Figure 7, to which you refer in your text on page 7. If the figure is no longer to be included as part of the submission please remove all reference to it within the text.

Reviewers' comments:

Reviewer's Responses to Questions

**Comments to the Author**

1. Is the manuscript technically sound, and do the data support the conclusions?

Reviewer #1: Partly

Reviewer #2: Yes

2. Has the statistical analysis been performed appropriately and rigorously? 

Reviewer #1: No

Reviewer #2: Yes

3. Have the authors made all data underlying the findings in their manuscript fully available?

Reviewer #1: Yes

Reviewer #2: Yes

4. Is the manuscript presented in an intelligible fashion and written in standard English?

Reviewer #1: Yes

Reviewer #2: Yes

5. Review Comments to the Author

Reviewer #1: ** What are the main claims of the paper and how significant are they for the discipline?

The paper develops a protocol for imaging of cerebral blood volume (CBV) weighted signals in the auditory cortex at 7T. This had not previously been attempted because of technical challenges that are specific to the auditory cortex, such as short arterial transit times and stronger sensitivity to cardiac pulsation.

The developed protocol attempts to overcome these limitations.

The work is significant since there are currently no other non-invasive technique capable of imaging CBVw signals in the auditory cortex. This advancement is important not only for the cognitive neuroscientific applications that the authors mention, but also to study the VASO signal contrast, which has implications for fMRI in general.

** Are the claims properly placed in the context of the previous literature? Have the authors treated the literature fairly?

Yes.

** Do the data and analyses fully support the claims? If not, what other evidence is required?

Partially. The protocol optimization section contains insightful details about crafting protocols specific for the auditory cortex, but it is debatable whether all of those optimatizations would generalize for other coils or other imaging resolutions - especially the highly specific reconstruction optimization. The claim that the protocol is ready to be used for the user base of application-focused neuroscientists is also not debatable, since the protocol hasn't actually been tested in any other settings/centers (at least according to the paper).

Also regarding VASO protocol optimization, the low auditory activation found after optimization is atributed to a sparse experimental task design, but readers may ask themselves whether that is truly the case, or if there is still something else going on with the sequence.

Beyond the sequence, and with focus on the auditory cortex, Scouten 2007 noted that VASO in the auditory cortex may have a large CSF signal contribution, though granted in their data they don't account for everything else that the current manuscript is accounting for (such as the short transit time), but CSF contamination could be a concern for readers (AFAIK the cited reference regarding magnetisation reset in VASO does not comment on fully adressing dynamic partial CSF). Perhaps another sentence or two explaining how magnetisation reset accounts for dynamic partial CSF would be beneficial.

Regarding the 2D vs 3D study, and the tonotopic mapping, the low number of subjects and the extremely low sensitivity of VASO (as seen by the low z-scores) are a potential concern. Even with BOLD studies in the visual/motor cortex one would be wary of making any conclusions from data with such small dataset / low z-scores.

But even assuming that the z-scores are reliable, the results on tonotopic mapping and cortical depth dependency would also vastly benefit from more data and/or analysis: the cortical depth profiles differ substantially between participants even after pooling data across layers, and the tonotopic maps from both subjects and both hemispheres are all different - no clear pattern can be seen in the VASO data. Attempts at tonotopic mapping would also imply that VASO has sufficient signal at or close to the voxel level to extract detailed information, but that cannot be concluded based on the results shown, and so it is hard to known whether the results correspond to inherent subject variability, or indeed something is still off with the sequence / analysis / experimental design.

What other evidence is required?

More control experiments would be extremely helpful. Simpler experiments to validate VASO, even if the conclusion is that VASO has low sensitivity, as mentioned in the discussion. More sanity checks that the signal follows a reproducible pattern are important.

Some potential ideas:

- Compare tonotopic maps from neighboring cortical depths. There should be some similarity.

- Show time-series of VASO and BOLD timeseries against the experimental paradigm and % signal changes. Are % signal changes negative upon activation?

- Compare tonotopic maps and cortical depth profiles against maps generated from resting-state data (perhaps even from experiment 2?).

- Show that cortical depth dependent profiles and tonotopic maps are reproducible in the same participant in two different sessions if data is or can be made available.

- monoaural stimulation? -> compare right and left hemisphere profiles. Some studies (see Gutschalk 2014) suggest that slow amplitude modulated noise should elicit mostly contralateral activation. Task could be used as a benchmark for task CNR. Auditory cortex is not my area of expertise, so maybe there are even simpler tasks that would do the job.

Of course, the suggestion is not to do all of the above, but to implement some form of sanity check that shows that the signals are reliable and could be interpreted as robust CBVw signals.

** PLOS ONE encourages authors to publish detailed protocols and algorithms as supporting information online. Do any particular methods used in the manuscript warrant such treatment? If a protocol is already provided, for example for a randomized controlled trial, are there any important deviations from it? If so, have the authors explained adequately why the deviations occurred?

The authors share all necessary information and are open about both the data and processing tools. No clear deviations.

** If the paper is considered unsuitable for publication in its present form, does the study itself show sufficient potential that the authors should be encouraged to resubmit a revised version?

Absolutely.

** Are original data deposited in appropriate repositories and accession/version numbers provided for genes, proteins, mutants, diseases, etc.?

Yes.

** Does the study conform to any relevant guidelines such as CONSORT, MIAME, QUORUM, STROBE, and the Fort Lauderdale agreement?

Yes.

** Are details of the methodology sufficient to allow the experiments to be reproduced?

Yes.

** Is any software created by the authors freely available?

Yes.

** Is the manuscript well organized and written clearly enough to be accessible to non-specialists?

Yes.

** Is it your opinion that this manuscript contains an NIH-defined experiment of Dual Use concern?

No.

Reviewer #2: In this manuscript, the authors presented a technical study to optimize the VASO fMRI pulse sequence for laminar fMRI in the human auditory cortex on 7T. The authors did a great job describing technical details to address particular challenges in this brain region and discussing potential advantages and limitations. I also like the open access to the imaging protocol which can be used and disseminated easily in the research community. The manuscript is well written and it is certainly of interest to readers. I have the following minor suggestions for the authors to consider:

1. The presentation of study 1 is a little strange to me. Normally, the methods and results should be in respective sections. As it is written now, the results of study 1 are also presented in the Methods section. While I understand that this is mainly because results from study 1 are needed to determine the parameters in studies 2 and 3, I still think that it is better to separate the methods and results for each study. For instance, one can say in methods for study 2 that the parameters used in this study are based on our results from study 1, see Results, etc. Alternatively, the authors can completely remove study 1, but instead just describe the optimized sequence and parameters. But I feel the later would reduce the technical importance of this work.

2. “When collecting simultaneous VASO and BOLD, these effects were more pronounced in the VASO data (Fig 1A).” Could the authors explain why the images in Fig. 1A support this statement. I cannot find it in the text or in the caption.

3. “Readout time longer than the cardiac cycle resulted in loss of contrast around Heschl’s gyrus (HG) and in typical vascular artifacts in components extracted with independent component analysis (ICA) from VASO time series (figure 1B)” Could the authors provide more details, for instance, what readout durations did they test, in how many subjects, other parameters, how was the ICA performed, how are the ICA components selected, why did Fig. 1B support this statement, etc. Again, I cannot find these information in the text or in the caption. I understand that some of these can go into the supplement. But I feel that these information is critical for the readers to fully appreciate this study.

4. “The segmented reference resulted in the best compromise between artifact level and tSNR in temporal areas.” I’d appreciate some quantitative results here if the authors choose to keep study 1 as one of the sub-studies.

6. PLOS authors have the option to publish the peer review history of their article (what does this mean?). If published, this will include your full peer review and any attached files.

Reviewer #1: No

Reviewer #2: No

---

## [Author Response · Author response to Decision Letter 0]

18 Nov 2022

We have uploaded a separate 'Response to Reviewers' file which contains a response to the reviewers and the editor. We copy here the text as requested by the system, but please note that figures are not uploaded here so please refer to the separate file. 

Comments of Editor (Ahveninen)

Journal Requirements:

All the requirements have been fulfilled.

2. Please upload a copy of Figure 7, to which you refer in your text on page 7. If the figure is no longer to be included as part of the submission please remove all reference to it within the text.

We thank the editor for this comment, but we believe that this stems from a misunderstanding. The original manuscript included five original figures, which were referred to as Fig 1-5. In page 7 of the original version of the manuscript (now page 11), we refer to figure 7 of an already published manuscript from authors Mildner et al., 2014. To avoid confusion, we have now rephrased the way we refer to these results:

“see reference [63], in particular the results reported in its Fig 7B”

We hope that in this way we have clarified that this figure has to be found in a separate paper.

This requirement has been fulfilled.

Reviewer 1

Reviewer #1: ** What are the main claims of the paper and how significant are they for the discipline?

The paper develops a protocol for imaging of cerebral blood volume (CBV) weighted signals in the auditory cortex at 7T. This had not previously been attempted because of technical challenges that are specific to the auditory cortex, such as short arterial transit times and stronger sensitivity to cardiac pulsation.

The developed protocol attempts to overcome these limitations.

The work is significant since there are currently no other non-invasive technique capable of imaging CBVw signals in the auditory cortex. This advancement is important not only for the cognitive neuroscientific applications that the authors mention, but also to study the VASO signal contrast, which has implications for fMRI in general.

We thank the reviewer for her/his comment in their appreciation of this work. In what follows we reply to each of the issues raised point by point. 

** Do the data and analyses fully support the claims? If not, what other evidence is required?

Partially. The protocol optimization section contains insightful details about crafting protocols specific for the auditory cortex, but it is debatable whether all of those optimatizations would generalize for other coils or other imaging resolutions - especially the highly specific reconstruction optimization. The claim that the protocol is ready to be used for the user base of application-focused neuroscientists is also not debatable, since the protocol hasn't actually been tested in any other settings/centers (at least according to the paper).

We thank the reviewer for the comment, which we agree with. In the revised version of the manuscript we have toned down the claims of generalizability of the developed protocol and have clarified what the main contribution of this work is - that is, the investigation of the challenges that developing a VASO protocol for the auditory has, and the strategies that we have followed to overcome them. 

In the abstract we have removed the claim of developing an optimized protocol for auditory neuroscientific applications, but clearly state that:

“We describe the main challenges we encountered when developing a VASO protocol for auditory neuroscientific applications and the mitigation strategies we have adopted.” 

In the introduction (page 4) we emphasize the exploratory aim of the study by tackling challenges we encountered and state that:

“Here, we present the results of the exploration of a wide parameter space aimed at mitigating methodological and physiological challenges encountered when using VASO to image the auditory cortex at submillimeter resolution.”

In the methods and results section we have rephrased any reference to an optimal protocol and in the discussion (page 18-19) we clearly state that:

“In this study, we aimed to develop a VASO protocol for laminar fMRI investigations of the auditory cortices by mitigating methodological and physiological challenges.”

We agree with the referee that all of these protocol optimizations might not generalize across a wider range of brain areas and hardware configurations. In the revised version of the manuscript we tone down our claims. Therefore, in the discussion (page 23) we toned down the significance of our work by stating that:

“We believe that the significance of this work is multi-fold. In study 1, we describe the approach we followed to tackle the main challenges encountered when using VASO for submillimeter auditory investigations. Following these steps may prove useful in case the resulting protocol we describe here would not generalize outside of the specific applications (as well as coils and imaging resolutions) we present. Nevertheless, we believe our results represent a first necessary step towards generalization as the protocol resulting from study 2 is made available for the user base of application-focused neuroscientists for testing in a wider range of application settings. The sequence binaries and the importable protocols are publicly available via ‘SIEMENS’ sequence ‘app-store’ on TEAMPLAY for any users of a ‘classical’ MAGNETOM 7T, which is the most widely used 7T scanner version around the world today. Users of other scanner versions and vendors can benefit from this protocol-development study as they can re-implement the acquisition approaches as described in study 1.”

We have removed any reference to an optimal protocol and specified that we do not suggest that this protocol can be used immediately in other challenging areas without any testing or changes. But we would like to note that the steps we have undertaken might help other researchers in their endeavors to image challenging areas. The discussion section of the manuscript contains the following vague outlook (without strong claims). 

“The imaging protocol developed here may have implications beyond the auditory cortex. The auditory cortex is not the only brain area challenged by proximal macro-vessels with substantial physiological noise. There are many other brain areas in which sub-millimeter VASO was not successfully applied until now, for example, hippocampus, insular cortex, claustrum, entorhinal cortex, and thalamic nuclei. Researchers investigating areas with similar artifacts could test whether following similar strategies might benefit them in these challenging areas. Finally, the main aim of this work was to provide the auditory research community with a viable VASO protocol for laminar fMRI studies, which is now available for testing by the community.”

We would like to note that the phrasing of the above mentioned paragraphs is not in contrast with the reviewer’s comment. We do not claim that the generalizability is not debatable. We only think that our increased understanding of potential signal artifacts and how to mitigate them may be informative to the process of developing a protocol for other brain areas. 

These paragraphs are motivated by feedback that we have gotten from three previous international conferences to abstracts of this work (Benelux ISMRM 2022, ISMRM London 2022, OHBM 2022). Following these comments, other labs have asked our advice when using a similar strategy to set up layer-fMRI protocols for medial-temporal cortex. 

Also regarding VASO protocol optimization, the low auditory activation found after optimization is atributed to a sparse experimental task design, but readers may ask themselves whether that is truly the case, or if there is still something else going on with the sequence.

We thank the reviewer for the comment which requires some clarification. 

In the original manuscript we referred to results obtained with a sparse design at the end of the description of study 1 and then in the discussion. The reason for this is that we thought it necessary to justify our choice to use continuous sound stimulation (study 2 and 3) in long blocks when this is not conventional in auditory fMRI (where sparse designs are commonly employed). When preparing the acquisitions of study 2 and 3, we followed the conventional use of long blocks in VASO studies. In choosing how to present sounds within these long blocks we initially followed a more “auditory conventional” sparse approach (900 ms in a TR of 3.4 seconds [that is a noise period of 2.5 seconds and a silent period of 900 ms in every TR]). These initial attempts, which we do not show, did not result in strong cortical responses. Following our past experience, in which we investigated the interplay between the length of silent gaps and scanner noise in eliciting auditory responses (De Martino et al., 2015), we reasoned that these weak activations could be due to the relatively low level of auditory stimulation compared to scanner noise. 

 To clarify this we have made several changes to the manuscript. Following the comment of reviewer 2, we decided to put the results of study 1 in the results section. This allowed us to present aur rationale for choosing a continuous stimulation paradigm in the methods section already (now page 8) and remove any reference to this issue from the discussion. We consider it reassuring that with this less conventional sound presentation approach, tonotopic maps resulting from the VASO (Fig 6) and BOLD acquisition (S4 Fig) conform to the expected topography.

 We hope these changes have clarified this issue.

For reference see: De Martino F, Moerel M, Ugurbil K, Formisano E, Yacoub E. Less noise, more activation: Multiband acquisition schemes for auditory functional MRI: Multiband Acquisition Schemes for Auditory fMRI. Magn Reson Med 2015;74:462–7. https://doi.org/10.1002/mrm.25408.

Beyond the sequence, and with focus on the auditory cortex, Scouten 2007 noted that VASO in the auditory cortex may have a large CSF signal contribution, though granted in their data they don't account for everything else that the current manuscript is accounting for (such as the short transit time), but CSF contamination could be a concern for readers (AFAIK the cited reference regarding magnetisation reset in VASO does not comment on fully adressing dynamic partial CSF). Perhaps another sentence or two explaining how magnetisation reset accounts for dynamic partial CSF would be beneficial.

We agree with the reviewer that traditional VASO methods can be significantly affected by dynamic CSF redistributions (Scouten 2007; Donahue 2006). We would like to note, however, that this is not the case for SS-SI VASO as it is for traditional VASO. Different from traditional steady-state VASO, in SS-SI VASO the blood nulling time is not based on steady-state blood magnetization, but it is based on once-inverted blood. Thus, in SS-SI VASO, the CSF z-magnetization is not negative (as it was for Scouten, Donnaue and others). In fact, in SS-SI VASO the CSF magnetization can be adjusted by the user to a desired value. Here, we employed an additional spin-reset pulse (Lu, 2008) of 70 deg in SS-SI VASO. This allowed us to maintain the CSF magnetization between the blood nulling and GM-magnetization. This is schematically depicted in the figure below: 

Furthermore we would like to note that we used tasks that engage a focal part of the brain (auditory cortex) only. This is different from systemic global tasks (like hypercapnia) used by Scouten and Constable. It is believed that these focal tasks do not affect CSF volume at a measurable level (Huber 2014b).

For more explanations on the different CSF sensitivity of SS-SI VASO compared to traditional VASO see also: 

● Empirical evidence for visual tasks in PhD thesis: https://drive.google.com/file/d/1OU5fUJHS87VCQPvVIbNPgVZynJI5wmUb/view?usp=sharing (see section 5.4.3 Changes in CSF volume starting at page 176).

● See discussion section in the original SS-SI VASO paper Huber 2014. See section “Dynamic Changes of CSF Volume” on page 8. 

In the revised version of the manuscript, we followed the reviewer's advice and added a few additional sentences that summarize the above explanations.

In the methods (page 14) we clarify that:

“The purpose of the reset pulse was also to effectively saturate stationary Mz-magnetization of cerebrospinal fluid (CSF) and gray matter (GM) before the application of the consecutive inversion pulse. At the blood nulling time in the subsequent TR, this results in a positive Mz-magnetization of CSF with a magnitude smaller than GM. Having a positive CSF Mz-magnetization in SS-SI-VASO is in contrast to the negative CSF magnetization in the traditional VASO approach. The suppressed CSF signal (see contrast in Fig 1E) mitigates potential biases of dynamic CSF volume changes that have previously been reported to impose a source of bias for VASO applications in the auditory cortex [75]. ”

In the discussion on page 19:

“Assuming the skull as a container of a fixed volume, increase of one compartment, e.g., CBV, must be compensated by volume decrease in another compartment, e.g., GM or CSF. VASO signal change is based on the idea that CBV increase is compensated by GM volume decrease only. However, depending on the brain region stimulated, a small dynamic change in CSF volume in the range of 0.5% (for neurally-induced tasks) to 10% (for systemic gas-breathing induced hypercapnia) has been experimentally observed (Scouten 2007; Donahue 2006). Such stimulus-dependent variations in CSF volume could cause an incorrect calculation of CBV changes from the VASO signal change (Scouten 2007; Donahue 2006, Jin 2010). In contrast with to CSF-nulled ACDC VASO (Scouten and Constable 2008) or VASO FLAIR (Donahue et al 2006) techniques, the SS-SI VASO signal changes in this study (with the employment of a 70 deg spin-reset pulse) reflect a positive CSF z-magnetization. Thus, the CBV change presented here reflects both components of the CBV change—the CBV increase that is compensated by a GM volume decrease as well as the CBV increase that is compensated by CSF volume decrease—with similar weighting.”

Since such CSF magnetization manipulations are widely adopted in the VASO community, and since are standardly applied in almost all VASO studies of the last decade, we refrained from including more explanations into the revised manuscript.

References: 

● Donahue MJ, Lu H, Jones CK, Edden RAE, Pekar JJ, van Zijl PCM. Theoretical and experimental investigation of the VASO contrast mechanism. Magn Reson Med 2006;56:1261–73. https://doi.org/10.1002/mrm.21072.

● Jin T, Kim SG. Change of the cerebrospinal fluid volume during brain activation investigated by T1ρ-weighted fMRI. NeuroImage. 2010 Jul;51(4):1378–83. 

● Huber et al., 2014, MRM “Slab-Selective, BOLD-Corrected VASO at 7 Tesla Provides Measures of Cerebral Blood Volume Reactivity with High Signal-to-Noise Ratio.”

● Huber L, Kennerley AJ, Gauthier CJ, Krieger SN, Maria Guidi DI, Turner R, et al. Cerebral blood volume redistribution during hypercapnia. Imaging Cerebral Physiology: Manipulating Magnetic Resonance Contrast through Respiratory Challenges. Talk presented at Leipzig Symposium 2014b;2:O4 https://doi.org/10.7490/f1000research.1115082.1

● Lu H. Magnetization “reset” for non-steady-state blood spins in Vascular-Space-Occupancy ( VASO ) fMRI. Proceedings of the 16th Annual Meeting ISMRM. 2008;16(1):2008. 

● Scouten A, Constable RT. Applications and limitations of whole-brain MAGIC VASO functional imaging. Magn Reson Med 2007;58:306–15. https://doi.org/10.1002/mrm.21273.

Regarding the 2D vs 3D study, and the tonotopic mapping, the low number of subjects and the extremely low sensitivity of VASO (as seen by the low z-scores) are a potential concern. Even with BOLD studies in the visual/motor cortex one would be wary of making any conclusions from data with such small dataset / low z-scores.

But even assuming that the z-scores are reliable, the results on tonotopic mapping and cortical depth dependency would also vastly benefit from more data and/or analysis: the cortical depth profiles differ substantially between participants even after pooling data across layers, and the tonotopic maps from both subjects and both hemispheres are all different - no clear pattern can be seen in the VASO data. Attempts at tonotopic mapping would also imply that VASO has sufficient signal at or close to the voxel level to extract detailed information, but that cannot be concluded based on the results shown, and so it is hard to known whether the results correspond to inherent subject variability, or indeed something is still off with the sequence / analysis / experimental design.

What other evidence is required?

More control experiments would be extremely helpful. Simpler experiments to validate VASO, even if the conclusion is that VASO has low sensitivity, as mentioned in the discussion. More sanity checks that the signal follows a reproducible pattern are important.

We thank the reviewer for the following suggestions and agree that our study could benefit from more reproducibility checks. Therefore, we followed some of the potential ideas expressed below. In particular, we have collected more data aimed at evaluating test-retest reliability. We did this by collecting (in one participant) enough data to compare results across two independent splits. In particular, we collected 12 runs of the 3D acquisition protocol. We divided these in two independent splits of six runs each. Each split thus includes more data than what we originally had in study 2 (three runs) and is more comparable to the length of acquisitions in study 3 (five runs). Yet collecting 6 runs of functional data is within a reasonable time span ~1 hour of scanning. 

We used the experimental stimulation paradigm used in study 2, to focus primarily on the replication of activation maps and layer profiles. These new results are presented in the revised manuscript after study 2 (new figure 5).

Below we will discuss the reviewer’s original suggestions and elaborate how we have incorporated these suggestions into the revised manuscript or the reason why we decided not to further investigate a specific suggestion. 

Some potential ideas:

- Compare tonotopic maps from neighboring cortical depths. There should be some similarity.

We thank the reviewer for this suggestion. Although there have been a few studies on the existence of frequency preference across cortical depths - columnarity should not be expected throughout the temporal lobe (see e.g. Moerel et al., J Neurosc (2018) and De Martino et al., PNAS (2015)). This may be true only for very limited portions of primary cortical areas. This makes the use of stability of tonotopy across depth not an optimal measure of data quality in our opinion. 

We believe that to showcase the reliability of the estimated auditory responses is sufficient to focus on the reproducibility of the cortical activations elicited by the broadband stimuli used in Study 2. The tonotopy data remain an initial showcase of the utility of VASO and more studies will be needed to dive into the information that VASO can provide to the understanding of frequency preference.

- Show time-series of VASO and BOLD timeseries against the experimental paradigm and % signal changes. Are % signal changes negative upon activation? 

We have followed this suggestion and now we present time courses for VASO in Fig. 2 and in the new Fig. 5. The time courses are presented in percent signal change, they allow evaluating the stability of the response across presentation blocks (trials) and are as expected negative in response to the stimulation. In figure 5 time courses are presented separately for the two independent splits and together with the F-maps (activation maps) obtained by analyzing the two independent splits. We believe these results highlight the stability of the cortical responses and reproducibility of the effects despite the low percent signal changes (which are expected in VASO).

New figure 2:

- Compare tonotopic maps and cortical depth profiles against maps generated from resting-state data (perhaps even from experiment 2?). 

We thank the reviewer for this suggestion. We present improved tonotopic maps in now figure 6 (figure 5 in the previous version). We have applied standard processing steps that we previously did not consider - that is, as we focus on overall topographic responses we have smoothed moderately the maps (see e.g. Moerel et al., 2014). We believe this improves the interpretation of the tonotopic maps compared to expectations set by the literature.

New figure 6:

- Show that cortical depth dependent profiles and tonotopic maps are

 reproducible in the same participant in two different sessions if data is or can be made available. 

We have followed this suggestion, we provide a proof of reproducibility in two independent splits. As clarified above, we do this on the experimental paradigm of study 2 and for this data we also provide independent estimation of the layer profiles in the two splits of the data (figure 5). 

New figure 5:

- monoaural stimulation? -> compare right and left hemisphere profiles. Some studies (see Gutschalk 2014) suggest that slow amplitude modulated noise should elicit mostly contralateral activation. Task could be used as a benchmark for task CNR. Auditory cortex is not my area of expertise, so maybe there are even simpler tasks that would do the job. 

We thank the reviewer for the suggestion. We believe that if reproducibility was to be assessed the better choice was (as suggested by the reviewer) to acquire the same data twice and compare results. The presentation of time courses (as suggested by the reviewer) in addition gives an idea of the stability of the response. The suggestion to use tasks (monaural stimulation or others) would in our opinion deserve a whole new publication. 

Of course, the suggestion is not to do all of the above, but to implement some form of sanity check that shows that the signals are reliable and could be interpreted as robust CBVw signals.

We hope that by presenting time courses and having done a reproducibility study we now present enough evidence of the robustness of the CBVw signal.

Reviewer 2

Reviewer #2: In this manuscript, the authors presented a technical study to optimize the VASO fMRI pulse sequence for laminar fMRI in the human auditory cortex on 7T. The authors did a great job describing technical details to address particular challenges in this brain region and discussing potential advantages and limitations. I also like the open access to the imaging protocol which can be used and disseminated easily in the research community. The manuscript is well written and it is certainly of interest to readers. I have the following minor suggestions for the authors to consider:

We thank the reviewer for her/his comment in their appreciation of this work. In what follows we reply to each of the issues raised point by point.

Before addressing all the specific comments, we would like to make one clarification about the nature of study 1. Study 1 is aimed at describing the procedures we followed to mitigate artifacts that we encountered during the development of our protocol. We did not aim to quantify if any of the chosen parameters results in statistically significantly better results compared to others. Furthermore, we did not aim to make ultimate claims about our protocol being superior to other protocols. That is, study 1 is a qualitative description of why and how we chose our protocol parameters to be used for later quantitative assessments of the usability (in study 2 and 3). We think of study 1 as being comparable to the conventional ’pilot scans’, that most neuroscience-focused studies perform in 1 to 3 participants to see if there is anything obviously wrong with the setup that needs rectification. In conventional publications, these kinds of qualitative pilot tests are not shared with the public.

We believe that by presenting the reasoning that we followed to derive an effective protocol will help the community understand the reasoning underlying our choices and (as we clarify) hopefully aid future applications of VASO that may have to follow similar steps. While the data shown in Fig. 1 are qualitative in nature, with questionable generalizability across other study setups, they give the reader an idea about our reasoning of why we focused our quantitative assessment in study 2 on the specific protocols at hand. 

1. The presentation of study 1 is a little strange to me. Normally, the methods and results should be in respective sections. As it is written now, the results of study 1 are also presented in the Methods section. While I understand that this is mainly because results from study 1 are needed to determine the parameters in studies 2 and 3, I still think that it is better to separate the methods and results for each study. For instance, one can say in methods for study 2 that the parameters used in this study are based on our results from study 1, see Results, etc. Alternatively, the authors can completely remove study 1, but instead just describe the optimized sequence and parameters. But I feel the later would reduce the technical importance of this work.

We agree with the reviewer and in the revised version, we have changed the structure of the manuscript and have included the results of study 1 in the results and we have taken these results out of the methods section. Additionally, we tried to make the purpose of study 1 clearer in the revised version of the manuscript by changing the description of Study 1 to: 

“Study 1: Protocol development in pilot experiments”

2. “When collecting simultaneous VASO and BOLD, these effects were more pronounced in the VASO data (Fig 1A).” Could the authors explain why the images in Fig. 1A support this statement. I cannot find it in the text or in the caption.

We thank the reviewer for this comment and we agree it should be clarified further. Therefore on page (11-12) we put clarify that:

“The inflow effects result in very bright vessels in both the BOLD and the VASO data. However, the ratio between the background signal and the signal from the vessels is higher in the BOLD data compared to the VASO data. This results in lower relative contrast between tissue types in the VASO data compared to the BOLD data (Fig 1A). ”

3. “Readout time longer than the cardiac cycle resulted in loss of contrast around Heschl’s gyrus (HG) and in typical vascular artifacts in components extracted with independent component analysis (ICA) from VASO time series (figure 1B)” Could the authors provide more details, for instance, what readout durations did they test, in how many subjects, other parameters, how was the ICA performed, how are the ICA components selected, why did Fig. 1B support this statement, etc. Again, I cannot find these information in the text or in the caption. I understand that some of these can go into the supplement. But I feel that these information is critical for the readers to fully appreciate this study.

We thank the reviewer for this comment and agree that some clarification is necessary. We have now included information on which readout times we have tested and added information on the independent component analysis (page 12): 

“Second, we explored the effect that the readout time (and its relationship to the cardiac cycle) has on VASO data in the temporal cortex. Initially, we used a readout time of about 1235 ms (one participant), which is longer than the cardiac cycle. Such a long readout time resulted in loss of contrast around Heschl’s gyrus (HG). Therefore, we opted for using a readout time shorter than the cardiac cycle (700 ms) in subsequent volunteers in all three studies, to mitigate this artifact. An additional independent component analysis (FSL MELODIC, 30 components) on the VASO time series (Fig 1B) collected with a readout longer than the cardiac cycle showed that the component with the largest variance was a typical vascular artifact centered on the large vessels in the auditory cortex. These two results exemplify the effect of physiological noise originating from the cardiac cycle after which we took the approach to shorten the readout time. ”

4. “The segmented reference resulted in the best compromise between artifact level and tSNR in temporal areas.” I’d appreciate some quantitative results here if the authors choose to keep study 1 as one of the sub-studies.

We agree with the reviewer that providing a more quantitative assessment and providing more background of our choice of scan parameters would be helpful. 

We focused on a combination of two quality metrics: 1) first and foremost the ghost level, and 2) the expected phase consistency between ACS data and time series data, with its implication on tSNR stability across experiments. 

We added a new row in Fig. 1C quantifying the ghost level as the signal ratio between auditory cortex and background. It can be seen that the FLEET ACS approach exhibits worse ghosting values compared to segmented and single-shot. This result made us refrain from using FLEET in the following experiments. Since the echo time and the phase evolution of single-shot GRAPPA reference lines are approximately three times longer/stronger for single-shot ACS compared to the segmented approach, we expected that this would mitigate intermittent ghosting across the time series. This is expected to result in stable tSNR values across protocols and participants. Thus, we decided to use the segmented approach for the remainder of the study.

We now clarify this in the text (page 13): 

“The FLEET ACS approach exhibited worse ghosting compared to single-shot and segmented in our experimental setup. Therefore we decided to refrain from using FLEET in the following experiments. Since the echo time and the phase evolution of single-shot GRAPPA reference lines are approximately three times longer/stronger for single-shot ACS compared to the segmented approach, we expected that using a segmented approach would mitigate intermittent ghosting across the time series. This is expected to result in stable tSNR values across protocols and participants. Thus, we decided to use the segmented approach for the remainder of the study as this is expected to be the best compromise between artifact level and tSNR in temporal areas.”.

---

## [Decision Letter · Decision Letter 1]

10 Jan 2023

Cerebral blood volume sensitive layer-fMRI in the human auditory cortex at 7T: Challenges and capabilities

PONE-D-22-23894R1

Dear Dr. Faes,

We’re pleased to inform you that your manuscript has been judged scientifically suitable for publication and will be formally accepted for publication once it meets all outstanding technical requirements.

Kind regards,

Jyrki Ahveninen

Academic Editor

PLOS ONE

Additional Editor Comments (optional):

Reviewers' comments:

Reviewer's Responses to Questions

**Comments to the Author**

1. If the authors have adequately addressed your comments raised in a previous round of review and you feel that this manuscript is now acceptable for publication, you may indicate that here to bypass the “Comments to the Author” section, enter your conflict of interest statement in the “Confidential to Editor” section, and submit your "Accept" recommendation.

Reviewer #1: All comments have been addressed

Reviewer #2: All comments have been addressed

2. Is the manuscript technically sound, and do the data support the conclusions?

Reviewer #1: Yes

Reviewer #2: Yes

3. Has the statistical analysis been performed appropriately and rigorously? 

Reviewer #1: Yes

Reviewer #2: Yes

4. Have the authors made all data underlying the findings in their manuscript fully available?

Reviewer #1: Yes

Reviewer #2: Yes

5. Is the manuscript presented in an intelligible fashion and written in standard English?

Reviewer #1: Yes

Reviewer #2: Yes

6. Review Comments to the Author

Reviewer #1: (No Response)

Reviewer #2: I thank the authors for carefully addressing all my previous comments. Congratulations on a nice work!

7. PLOS authors have the option to publish the peer review history of their article (what does this mean?). If published, this will include your full peer review and any attached files.

Reviewer #1: No

Reviewer #2: No

---

## [Editor Report · Acceptance letter]

27 Jan 2023

PONE-D-22-23894R1 

Cerebral blood volume sensitive layer-fMRI in the human auditory cortex at 7T: Challenges and capabilities 

Dear Dr. Faes:

I'm pleased to inform you that your manuscript has been deemed suitable for publication in PLOS ONE. Congratulations! Your manuscript is now with our production department. 

Kind regards, 

on behalf of

Dr. Jyrki Ahveninen 

Academic Editor

PLOS ONE